

# The relationship between cancer associated fibroblasts biomarkers and prognosis of breast cancer: a systematic review and meta-analysis

Meimei Cui[1], Hao Dong[1], Wanli Duan[1], Xuejie Wang[1], Yongping Liu[1], Lihong Shi[2] and Baogang Zhang[1]

[1] Department of Pathology, School of Basic Medical Sciences, Shandong Second Medical University, Weifang, Shandong, China
[2] School of Rehabilitation Medicine, Shandong Second Medical University, Weifang, Shandong, China

## ABSTRACT

**Background**. To elucidate the relationship between cancer-associated fibroblast (CAFs) biomarkers and the prognosis of breast cancer patients for individualized CAFs-targeting treatment.

**Methodology**. PubMed, Web of Science, Cochrane, and Embase databases were searched for CAFs-related studies of breast cancer patients from their inception to September, 2023. Meta-analysis was performed using R 4.2.2 software. Sensitivity analyses were performed to explore the sources of heterogeneity. Funnel plot and Egger's test were used to assess the publication bias.

**Results**. Twenty-seven studies including 6,830 patients were selected. Univariate analysis showed that high expression of platelet-derived growth factor receptor-$\beta$ (PDGFR-$\beta$) ($P = 0.0055$), tissue inhibitor of metalloproteinase-2 (TIMP-2) ($P < 0.0001$), matrix metalloproteinase (MMP) 9 ($P < 0.0001$), MMP 11 ($P < 0.0001$) and MMP 13 ($P = 0.0009$) in CAFs were correlated with reduced recurrence-free survival (RFS)/disease-free survival (DFS)/metastasis-free survival (MFS)/event-free survival (EFS) respectively. Multivariate analysis showed that high expression of $\alpha$-smooth muscle actin ($\alpha$-SMA) ($P = 0.0002$), podoplanin (PDPN) ($P = 0.0008$), and PDGFR-$\beta$ ($P = 0.0470$) in CAFs was associated with reduced RFS/DFS/MFS/EFS respectively. Furthermore, PDPN and PDGFR-$\beta$ expression in CAFs of poorly differentiated breast cancer patients were higher than that of patients with relatively better differentiated breast cancer. In addition, there is a positive correlation between the expression of PDPN and human epidermal growth factor receptor-2 (HER-2).

**Conclusions**. The high expression of $\alpha$-SMA, PDPN, PDGFR-$\beta$ in CAFs leads to worse clinical outcomes in breast cancer, indicating their roles as prognostic biomarkers and potential therapeutic targets.

Corresponding authors
Lihong Shi, slh2020@wfmc.edu.cn
Baogang Zhang,
zbg0903@hotmail.com

## INTRODUCTION

Breast cancer has now surpassed lung cancer as the leading cause of global cancer incidence in 2020 and it is the fifth leading cause of cancer mortality worldwide (*Sung et al., 2021*). Identifying new predictive molecular biomarkers for progression and recurrence of cancer could promote diagnostic and therapeutic techniques and thus improve the survival of patients (*So et al., 2022*).

There is growing recognition that tumor growth depends not only on the malignant cancer cells themselves but also on the tumor microenvironment (TME) (*Li, Sun & Hu, 2017*). Fibroblasts are the most common stromal cell type of solid tumors and they are referred to as cancer-associated fibroblasts (CAFs) (*Ma et al., 2023*). A growing list of biomarkers, *e.g.*, $\alpha$-smooth muscle actin ($\alpha$-SMA), S100A4/fibroblast specific protein 1 (FSP-1), fibroblast activation protein (FAP), *etc.*, have been used to define activated CAFs (*Hu et al., 2022*) and CAFs are a potential therapeutic target for cancer treatment. On the other hand, it has been recognized that CAFs constitute heterogeneous subpopulations with distinct molecular characteristics (*Cully, 2018*; *Davidson et al., 2021*; *Sebastian et al., 2020*; *Sidaway, 2018*) and the relative function of specific biomarkers is likely to vary by tumor type and has yet to be defined fully, *e.g.*, the impact of podoplanin (PDPN) on the function of CAFs is controversial for breast cancer and lung cancer (*Takahashi et al., 2015*), the impact of different CAF subtypes on patient outcomes is a topic worth studying.

Until now, most researches on CAFs biomarkers for breast cancer have been limited to a few pre-determined biomarkers (*Hu et al., 2021*; *Hu et al., 2018a*; *Hu et al., 2018b*) and comprehensive analysis on the prognostic values of CAFs biomarkers in breast cancer has not been reported. This study conducted a systematic analysis to assess the relationship between CAFs biomarkers and the prognosis of breast cancers to provide more depth insights and offer stronger support for tailored individualized therapy.

## SURVEY METHODOLOGY

This meta-analysis was conducted in accordance with the Preferred Reporting Items for Systematic Reviews and Meta-analyses reporting guideline (*Moher et al., 2009*). The corresponding checklists was shown in the File S1. The study protocol was registered with the PROSPERO International Prospective Register of Systematic Reviews (CRD42023446096).

### Search strategy

We searched PubMed, Web of Science, Embase and Cochrane library from inception to September, 2023, using the medical subject headings ''cancer-associated fibroblasts'' and ''breast cancer''. There was no restrictions on publication format or language. The full search strategy for each database is available in File S2.

### Inclusion and exclusion criteria

Studies were included if they met the following criteria: (1) Studies on the relationship between CAFs and survival of breast cancer patients from inception to September, 2023; (2) CAFs biomarkers were detected by immunohistochemistry (IHC) in human breast

cancer specimens; (3) Provided hazard ratios (HRs) with 95% confidence intervals (CIs), or Kaplan–Meier (KM) curves of high and low biomarker-positive fibroblast density with survival outcomes. The following survival outcomes were evaluated: overall survival (OS), disease-specific survival (DSS), event-free survival (EFS), recurrence-free survival (RFS), disease-free survival (DFS), and metastasis-free survival (MFS).

The exclusion criteria were as follows: (1) review or full text including comments, case reports, letters to the editor, and conference abstracts; (2) insufficient data to estimate HRs; (3) fibroblasts detected without a biomarker or biomarker-positive staining in the tumor stroma or TME; (4) no cut-offs provided for low/high or negative/positive expression of biomarkers in CAFs in the results or methods section; (5) duplicate data or no full text available; (6) biomarkers appeared in less than two independent queues; (7) quality score of the study <6.

## Data extraction

Three researchers (MMC, HD and WLD) independently conducted literature screening, data extraction, and quality assessment and cross-checked each other's work. Disputes were resolved through consultation, or discussion with a third author (BGZ). The extraction content included the following information: first author, publication year, number of patients, median age, follow-up time, methods used to quantify fibroblasts, cut-off values used to determine the density of these cells, data on OS, DFS, DSS, RFS, EFS and MFS and clinical pathological features, including primary tumor, lymph nodes, distant metastasis (TNM) staging, human epidermal growth factor receptor-2 (HER-2), and tumor differentiation, from text, tables, and KM curves, and so on. If the study only provided a KM curve without relevant HRs, we used Engauge Digitizer 4.1 to extract the missing HRs from the survival curve data and analyzed using the tools developed by *Tierney et al. (2007)*.

## Literature quality score

Two authors (MMC and HD) independently conducted a literature quality assessment using the Newcastle-Ottawa Scale (NOS) (*Stang, 2010*) and reached a consensus with a third or more authors (YPL and LHS). The score of 6 or higher was considered high quality.

## Statistical analysis

After extracting all relevant data, we first grouped similar survival outcomes into two categories: OS/DSS, RFS/DFS/MFS/EFS. However, as suggested by *Riley et al. (2019)*, we considered both univariate and multivariate analysis-derived HRs separately. Using the inverse variance method, we created weighted HRs with 95% CIs and $P$ for random-effects models, based on at least two different cohorts of individual biomarkers with the same outcome group and analysis method. Forest plots for individual biomarkers were generated using R 4.2.2 (College Station, Texas). Heterogeneity was assessed by calculating $I^2$ and $\tau^2$ values, and $P$ were generated to evaluate the statistical significance of heterogeneity. For $I^2$ index, where $I^2 < 50\%$ indicated low heterogeneity among studies and a fixed-effects model was applied, while $I^2 > 50\%$ indicated high heterogeneity and a random-effects model was adopted (*Barendregt et al., 2013*; *Higgins & Green, 2008*). Statistical significance

was set at $P <0.05$. Subgroup and sensitivity analyses were performed to explore sources of heterogeneity. Publication bias in the meta-analysis was detected qualitatively by visual inspection of funnel plots and quantitatively by the Egger linear regression test (*Egger et al., 1997*; *Higgins & Green, 2008*).

## RESULTS

### Study characteristics

Among the 8,056 studies, 290 passed the screening based on title and abstract among which 27 studies were selected for systematic review and meta-analysis (*Amornsupak et al., 2017*; *Ariga et al., 2001*; *Busch et al., 2012*; *Cai et al., 2017*; *Egeland et al., 2017*; *Eiro et al., 2019*; *Eiró et al., 2015*; *Gonzalez et al., 2009*; *Jung, Lee & Koo, 2015*; *Kim, Lee & Koo, 2016*; *Martinez et al., 2015*; *Min et al., 2013*; *Muchlińska et al., 2022*; *Park, Jung & Koo, 2016*; *Park, Kim & Koo, 2015*; *Paulsson et al., 2009*; *Pula et al., 2011*; *Pula et al., 2013a*; *Pula et al., 2013b*; *Schoppmann et al., 2012*; *Strell et al., 2019*; *Surowiak et al., 2007*; *Tanaka et al., 2021*; *Yamashita et al., 2012*; *Yang et al., 2017*; *Zhang et al., 2008*; *Zhou et al., 2018*) (Fig. 1). Among the 27 included studies, nine identified protein biomarkers appeared in at least two independent cohorts, allowing for meta-analysis. The earliest included study was published in 2001, however, the majority (18/27, 66.67%) were published within the past 10 years, which can reflect the increasing interest in CAFs. The cohort size ranged from 16 to 642 individuals. Overall, 27 studies yielded a total of 69 survival outcome measures (Table S1).

### Correlation between the expression levels of CAFs biomakers and survival outcome

To comprehensively explore the relationship between CAFs biomarkers and the prognosis of breast cancer patients, meta-analysis for multiple independent studies were conducted. Univariate analysis revealed that higher expression of platelet-derived growth factor receptor-$\beta$ (PDGFR-$\beta$) (HR =1.51, 95% CI [1.13–2.03], $P = 0.0055$), tissue inhibitor of metalloproteinase-2 (TIMP-2) (HR = 5.50, 95% CI [3.66–8.27], $P <0.0001$), matrix metalloproteinase (MMP) 9 (HR = 3.42, 95% CI [2.22–5.28], $P <0.0001$), MMP 11 (HR = 2.70, 95% CI [1.50–4.86], $P = 0.0009$), and MMP 13 (HR = 2.44, 95% CI [1.31–4.56], $P = 0.0052$) in CAFs were associated with shorter RFS/DFS/MFS/EFS respectively. In addition, multivariate analysis revealed that higher expression of α-SMA (HR = 2.79, 95% CI [1.62–4.83], $P = 0.0002$), PDPN (HR = 2.57, 95% CI [1.48–4.46], $P = 0.0008$) and PDGFR-$\beta$ (HR = 1.40, 95% CI [1.00–1.96], $P = 0.0470$) in CAFs were associated with shorter RFS/DFS/MFS/EFS of breast cancer patients respectively (Table 1).

Moreover, we found that the expression of PDPN and PDGFR-$\beta$ in CAFs were associated with histological grade of breast cancers and high PDPN ($P = 0.0339$) or PDGFR-$\beta$ ($P < 0.0001$) expression happened in poorly differentiated breast cancer tissues respectively. Furthermore, there was significantly higher PDPN expression in CAFs of HER-2-positive breast cancers ($P = 0.0095$) (Fig. 2).
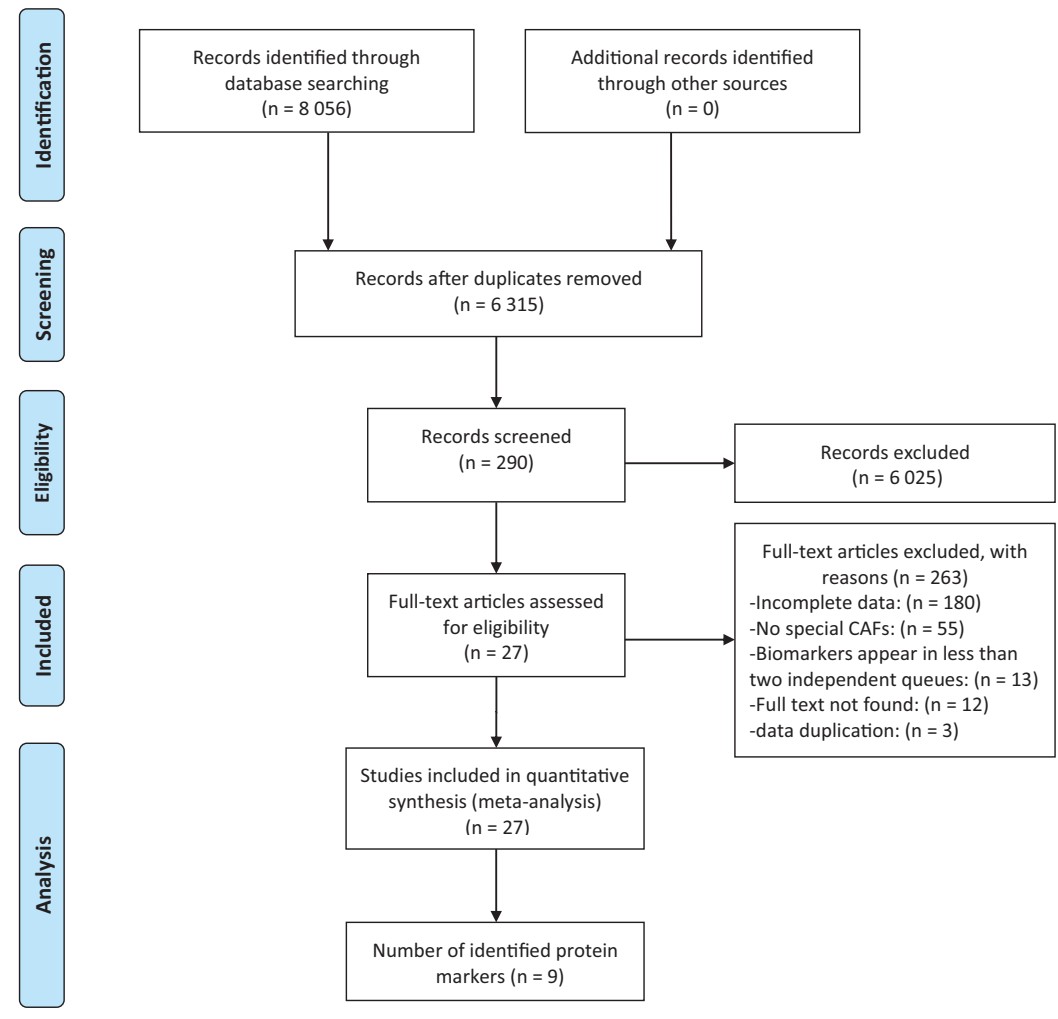

**Figure 1** Flow chart describing steps carried out in selecting articles.

## Sensitivity analysis

To validate the reliability of the results, sensitivity analysis was conducted to ensure the robustness of the study. The results showed that each individual study did not significantly impact the overall outcomes for RFS/DFS/MFS/EFS, indicating the results of this study were reliable (Fig. S1–S4).

## Publication bias

To enhance the credibility of our conclusions, publication bias analysis was conducted using funnel plots and Egger's test. Funnel plot and Egger's tests indicated that no potential publication bias existed between CAFs biomarkers and OS/DSS ($P > 0.05$) or RFS/DFS/MFS/EFS ($P > 0.05$) (Fig. S5).

**Table 1  Summary of results from the random effects models for individual markers.**

| Marker | Outcomes | Analysis | Studies | Random effects model | | Heterogeneity | | |
|---|---|---|---|---|---|---|---|---|
| | | | | Overall effect (95% CIs) | P | I² (%) | τ² | P |
| α-SMA | OS/DSS | Univariate | 4 | 1.14 (0.36 - 3.56) | 0.8264 | 62.8 | 0.82 | 0.0448 |
| | RFS/DFS/MFS/EFS | Univariate | 4 | 2.30 (0.53 - 9.94) | 0.2654 | 67.7 | 1.55 | 0.0258 |
| | | Multivariate | 3 | 2.79 (1.62 - 4.83) | 0.0002 | 0.0 | <0.0001 | 0.5019 |
| PDPN | OS/DSS | Univariate | 5 | 1.38 (0.31 - 6.19) | 0.6769 | 82.2 | 2.43 | 0.0002 |
| | | Multivariate | 5 | 1.94 (0.79 - 4.78) | 0.1476 | 63.7 | 0.63 | 0.0263 |
| | RFS/DFS/MFS/EFS | Univariate | 3 | 0.79 (0.20 - 3.13) | 0.7362 | 89.4 | 1.34 | <0.0010 |
| | | Multivariate | 3 | 2.57 (1.48 - 4.46) | 0.0008 | 32.8 | 0.09 | 0.2249 |
| PDGFR-β | OS/DSS | Univariate | 3 | 1.44 (0.46 - 4.48) | 0.5286 | 0.0 | 0.00 | 0.8347 |
| | RFS/DFS/MFS/EFS | Univariate | 4 | 1.51 (1.13 - 2.03) | 0.0055 | 0.0 | 0.00 | 0.9328 |
| | | Multivariate | 3 | 1.40 (1.00 - 1.96) | 0.0470 | 0.0 | 0.00 | 0.9768 |
| FAP | OS/DSS | Multivariate | 2 | 0.81 (0.05 - 12.27) | 0.8808 | 93.4 | 3.59 | 0.0001 |
| | RFS/DFS/MFS/EFS | Multivariate | 2 | 0.89 (0.11 - 7.10) | 0.9126 | 94.6 | 2.13 | <0.0001 |
| FSP-1 | OS/DSS | Univariate | 5 | 0.67 (0.31 - 1.45) | 0.3139 | 81.4 | 0.53 | 0.0002 |
| | RFS/DFS/MFS/EFS | Univariate | 3 | 0.99 (0.33 - 2.98) | 0.9895 | 86.9 | 0.81 | 0.0005 |
| TIMP-2 | RFS/DFS/MFS/EFS | Univariate | 3 | 5.50 (3.66 - 8.27) | <0.0001 | 0.0 | 0.00 | 0.7157 |
| MMP9 | RFS/DFS/MFS/EFS | Univariate | 3 | 3.42 (2.22 - 5.28) | <0.0001 | 0.0 | 0.00 | 0.9310 |
| MMP11 | RFS/DFS/MFS/EFS | Univariate | 4 | 3.18 (2.06 - 4.90) | <0.0001 | 62.2 | 0.11 | 0.0474 |
| MMP13 | RFS/DFS/MFS/EFS | Univariate | 3 | 1.98 (1.32 - 2.96) | 0.0009 | 0.0 | 0.00 | 0.6540 |

**Notes.**

α-smooth muscle actin = α-SMA; podoplanin = PDPN; fibroblast activation protein = FAP; fibroblast-specific protein-1 = FSP-1; platelet-derived growth factor receptor = PDGFR; tissue inhibitors of metalloproteinase = TIMP; matrix metalloproteinase = MMP; overall survival = OS; disease-specific survival = DSS; recurrence-free survival = RFS; disease-free survival = DFS; metastasis-free survival = MFS; confidence intervals = CIs.

## DISCUSSION

CAFs constitute a specific cell type within the TME and it is widely accepted that CAFs can facilitate tumor progression, promoting invasiveness and metastasis and diminishing survival rates of tumor patients mainly due to the "desmoplastic reaction" of CAFs (*Bochet et al., 2013*; *Tomasek et al., 2002*). The desmoplastic TME not only promotes malignant behaviors of cancer cells, but also prevents the entry of immune cells and drugs (*Cukierman & Bassi, 2010*). However, till now, many challenges in defining the origins, biomarkers and functions of CAFs still persist.

This study, employing a meta-analysis approach, reveals that the high expression of α-SMA, PDPN and PDGFR-β in CAFs may contribute to an unfavorable prognosis of breast cancer patients. Furthermore, the high expression of PDPN and PDGFR-β in CAFs is significantly correlated with poor differentiation of breast cancer. Moreover, a significant correlation is found between the high expression of PDPN in CAFs and HER-2-positive breast cancers.

As a skeletal protein, α-SMA is the most extensively used biomarkers of CAFs and is related to transforming growth factor-beta (TGF-β) production and high contractility of cancer cells (*Kunz-Schughart & Knuechel, 2002*; *Yoshida, 2020*). In breast cancer, the proportion of α-SMA⁺-CAFs was positively correlated with the proliferation, invasion, metastasis, and chemoresistance of tumor cells and negatively correlated with survival

### A. PDPN
#### Histological grade (grade 3/grade 1-2)

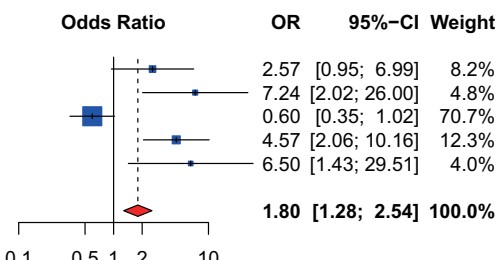

### B. PDPN
#### HER-2 (positive/negative)

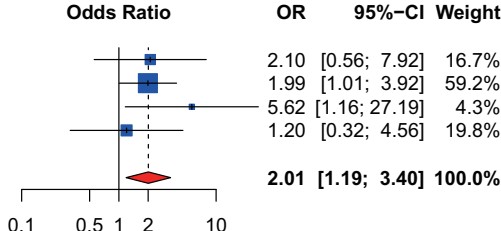

### C. PDGFR-β
#### Histological grade (grade 3/grade 1-2)

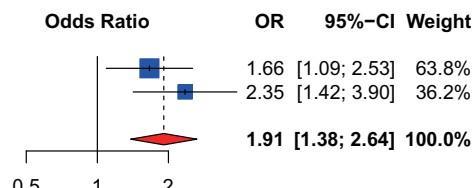

**Figure 2** **Forest plots indicating the association between the expression of PDPN, PDGFR-β in CAFs and histopathological feature of breast cancers.** (A) PDPN expression and histological grade. (B) the correlation between PDPN and HER-2 expression. (C) PDGFR-β expression and histological grade. *Strell et al. (2019)*; *Paulsson et al. (2009)*; *Pula et al. (2013a)*; *Pula et al. (2013b)*; *Schoppmann et al. (2012)*; *Pula et al. (2011)*; *Tanaka et al. (2021)*.

period. Although CAF-targeted nanoparticles for remodeling the TME has indeed reduced levels of $\alpha$-SMA expression and inhibited tumor growth(*Alili et al., 2011*), as $\alpha$-SMA is also expressed in other cell types, targeting $\alpha$-SMA$^+$-CAF has not yet achieved ideal results.

PDPN is a small transmembrane mucin-like glycoprotein that was initially characterized as a platelet-aggregation factor in cancer cells from colorectal tumors (*Kato et al., 2003*; *Quintanilla et al., 2019*). PDPN plays crucial functions in lymphangiogenesis (*Astarita, Acton & Turley, 2012*; *Renart et al., 2015*), cancer invasiveness, extracellular matrix (ECM) remodeling (*Hoshino et al., 2011*; *Ito et al., 2012*) as well as promoting an immunosuppressive microenvironment (*Sakai et al., 2018*). For breast cancer, PDPN$^+$-CAFs tended to result in a more malignant pathological status and could facilitate
immunosuppression and disease progression, which is consistent with the finding of this meta-analysis study and the findings of PDPN functions in multiple types of solid tumors (*Du et al., 2023*; *Yamaguchi et al., 2021*). Furthermore, recent research by *Du et al. (2023)* suggests that PDPN$^+$-CAFs contribute to HER-2-positive breast cancer resistance to trastuzumab by inhibiting antibody-dependent NK cell-mediated cytotoxicity. This study revealed that the high expression of PDPN in CAFs was associated with histological grade and HER-2 status, indicating that PDPN has clear potential as a cancer biomarker and therapeutic target. However, the underlying mechanism about the interaction and relationship between PDPN$^+$-CAFs and HER-2-positive breast cancer still needs for further investigation.

PDGFR (PDGFR-$\alpha$, PDGFR-$\beta$) is a tyrosine kinase receptor located on the surface of fibroblasts, neural precursor cells, astrocytes and pericytes. The expression of PDGFR-$\beta$ in CAFs was positively correlated with the poorly differentiated breast cancer tissues and poor prognosis. PDGFR-$\beta$ participates in fibroblast activation and transformation (*Jansson et al., 2018*; *Primac et al., 2019*) while inhibition of PDGFR signaling could transform CAFs into resting fibroblasts and inhibit angiogenesis and tumor growth (*Hu et al., 2022*). In addition, PDGFR-$\alpha$/$\beta$-positive CAFs induced the migration and M2 polarization of macrophage, thus modulating the immune microenvironment. Moreover, PDGFR-$\beta$ is considered as a key regulatory molecule for tumor drug resistance for high expression of stromal PDGFR-$\beta$ in breast cancer is associated with reduced benefit of tamoxifen (*Paulsson et al., 2017*). Interestingly, blocking of stromal PDGFR indeed reduced the interstitial fluid pressure, increased tumor drug uptake and enhanced the therapeutic efficacy of systemically delivered drugs (*Reed & Rubin, 2010*). Combining with the findings of this meta-analysis that PDGFR-$\beta^+$-CAFs is significantly correlated with poor differentiation and poor prognosis of breast cancer, targeting PDGFR pathways may be a potentially effective tumor treatment strategy.

Notably, although great progress has been achieved about the roles of CAFs biomarkers in breast cancer, due to dynamic expression pattern of biomarkers in CAFs during the progression of breast cancer (*Friedman et al., 2020*), even somewhat deficiency in specificity and sensitivity of CAFs biomarkers, much effort in translating the findings from bench to bedside is still needed. In addition, this study was conducted using both univariate and multivariate analyses, representing a more reliable approach compared to prior research, however, because of the limitation of published literature and data, further exploration with a larger sample size is necessary.

It should also be pointed out that in this study, univariate analysis indicates a correlation between TIMP-2 and MMP 9/11/13 and adverse prognosis of breast cancer patients, but considering the omission of other potential factors and the possible presence of spurious or indirect correlations in univariate analysis, the reliability of these findings is relatively lower. Moreover, it is susceptible to the influence of collinearity among independent variables (*Huberty & Morris, 1989*; *Trikalinos, Hoaglin & Schmid, 2014*). Therefore, the results from the univariate analysis are not considered and the outcomes of the multivariate analysis are interpreted robustly.

## CONCLUSIONS

In conclusion, the high expression of $\alpha$-SMA, PDPN, PDGFR-$\beta$ in CAFs led to unfavorable clinical outcomes in breast cancer patients, implicating that all these biomarkers could have potential values in the treatment and prognostic evaluation of breast cancer patients. Therefore, CAFs research, although challenges remain, will facilitate tailored targeting of CAFs, if not alone, then as combination strategy to optimize clinical benefits of breast cancers.

## ACKNOWLEDGEMENTS

Thank you to Pro LM Luo for the initial comments on the paper and for editing the final draft.

### Funding

This work was supported by the National Natural Scientific Foundation of China [No. 82373124, 81872163 and 81672631], the Weifang Science and Technology Development Program [No. 2021YX045, 2021YX072 and 2023YX043] and the Scientific and Technological Innovation Team [No. 02141607]. The funders had no role in study design, data collection and analysis, decision to publish, or preparation of the manuscript.

### Grant Disclosures

The following grant information was disclosed by the authors:
The National Natural Scientific Foundation of China: 82373124, 81872163, 81672631.
the Weifang Science and Technology Development Program: 2021YX045, 2021YX072, 2023YX043.
Scientific and Technological Innovation Team: No. 02141607.

### Competing Interests

The authors declare there are no competing interests.

### Author Contributions

- Meimei Cui conceived and designed the experiments, analyzed the data, authored or reviewed drafts of the article, and approved the final draft.
- Hao Dong performed the experiments, prepared figures and/or tables, and approved the final draft.
- Wanli Duan performed the experiments, prepared figures and/or tables, and approved the final draft.
- Xuejie Wang performed the experiments, prepared figures and/or tables, and approved the final draft.
- Yongping Liu analyzed the data, authored or reviewed drafts of the article, and approved the final draft.

- Lihong Shi conceived and designed the experiments, analyzed the data, authored or reviewed drafts of the article, and approved the final draft.
- Baogang Zhang conceived and designed the experiments, authored or reviewed drafts of the article, and approved the final draft.

## Data Availability

The raw data is available in the Supplementary Files.

## Supplemental Information

Supplemental information for this article can be found online at http://dx.doi.org/10.7717/peerj.16958#supplemental-information.

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
