# Peer review of "The relationship between cancer associated fibroblasts biomarkers and prognosis of breast cancer: a systematic review and meta-analysis"

_PeerJ, doi:10.7717/peerj.16958_

## Round 0.1 · original submission · Major Revisions

The manuscript suffers from a lack of specificity in the discussion and conclusions (abstracts). In the conclusions (abstract), the results (related factors) were not directly mentioned, which means the names of the parameters were not listed. Similarly, the discussion section does not have any analyses or connection with the results of this manuscript. This is a major issue. The introduction is another area that needs significant effort to update. The concepts and paragraphs are not well developed in that section.

Based on the reviewers' comments and my observations, I am convinced that this manuscript needs major changes to make it ready for reconsideration. Please ensure that you address all the comments from the reviewer. Thank you

Reviewer 1 ·

Basic reporting

This is a very informative review, which addresses an important question related to the importance of CAFs for diagnosis/prognosis of breast cancer patients.
The review is well written, the problem was well introduced
Comments:
1. english: needs revision of several words and also to shorten some long and non-clear sentences, such as in the introduction lane 55-60 as well as the last sentence in the conclusion.

2. The discussion need improvement, the present discussion is a simple repetition of the introduction and the results without deep insight on the importance of all these genes in the breast cancer prognosis and how they may be used for targeted therapy specifically in CAFs, because several of these genes such as a-SMA is also expressed in myoepithelial cells.

Experimental design

No comment

Validity of the findings

No comment

Additional comments

No additional comment

Reviewer 2 ·

Basic reporting

This systematic review and meta-analysis by Cui Meimei et al. aimed to elucidate the relationship between CAF biomarkers and the prognosis of breast cancer patients, focusing on individualized CAF-targeting treatment. The study utilized various databases to gather relevant studies, included 27 studies with 6,830 patients, examining the expression of various biomarkers in CAFs and their correlation with various survival outcomes. The study found that high expression of certain biomarkers such as PDGFR-β, TIMP-2, MMP 9, MMP 11, and MMP 13 in CAFs were correlated with reduced survival outcomes. They highlighted the potential use of these biomarkers in tailoring CAF-targeted therapies and prognostic evaluations in breast cancer patients.

Experimental design

1. The study does not thoroughly explore the heterogeneity in biomarker expression among different CAF subtypes and its implications for breast cancer prognosis.
2. The study does not sufficiently address the contradictory findings reported in the literature regarding the prognostic value of certain CAF biomarkers, which could be due to the functional and phenotypic heterogeneity of the tumor microenvironment.
3. The study employs both univariate and multivariate analyses, there is a lack of depth in statistical exploration, particularly in addressing the complex interactions between different biomarkers and clinical outcomes.

Validity of the findings

no comment

---

## Round 0.2 · accepted · Accept

Thank you for addressing the reviewer comments. Congratulations!

Reviewer 1 ·

Basic reporting

Clear and unambiguous, professional English used throughout

Experimental design

Research question well defined, relevant & meaningful

Validity of the findings

NA

Additional comments

Very good and informative review on a very important subject

Reviewer 2 ·

Basic reporting

The revised version meets my expectations and aligns well with the requirements for publication in this journal.

Experimental design

no comments.

Validity of the findings

no comments.